# Changes in Racial/Ethnic Disparities in Opioid-Related Outcomes in Urban Areas during the COVID-19 Pandemic: A Rapid Review of the Literature

**DOI:** 10.3390/ijerph19159283

**Published:** 2022-07-29

**Authors:** Kara M. Moran, Pricila H. Mullachery, Stephen Lankenau, Usama Bilal

**Affiliations:** 1College of Nursing and Health Professions, Drexel University, 1601 Cherry Street, Philadelphia, PA 19102, USA; kmm848@drexel.edu; 2Urban Health Collaborative, Dornsife School of Public Health, Drexel University, 3600 Market Street, Philadelphia, PA 19104, USA; ub45@drexel.edu; 3Department of Community Health and Prevention, Dornsife School of Public Health, Drexel University, 3215 Market Street, Philadelphia, PA 19104, USA; sel59@drexel.edu; 4Department of Epidemiology and Biostatistics, Dornsife School of Public Health, Drexel University, 3215 Market Street, Philadelphia, PA 19104, USA

**Keywords:** opioid use disorders, health disparities, urban health, COVID-19

## Abstract

Opioid use disorders (OUDs) are increasingly common among minoritized populations, who have historically experienced limited access to healthcare, a situation that may have worsened during the COVID-19 pandemic. Using a structured keyword search in Pubmed, we reviewed the literature to synthesize the evidence on changes in racial/ethnic disparities in OUD-related outcomes in urban areas during the COVID-19 pandemic in the US. Nine articles were included in the final analysis. Six found increases in OUD-related outcomes during the pandemic, with four showing a widening of disparities. Results also point to the worsening of opioid outcomes among Black and Latinx individuals related to shelter-in-place or stay-at-home orders. Studies examining the use of telehealth and access to OUD treatment showed that minoritized groups have benefited from telehealth programs. The limited number of studies in a small number of jurisdictions indicate a gap in research examining the intersection between COVID-19 and OUD-related outcomes with a focus on disparities. More research is needed to understand the impact of the COVID-19 pandemic and related policies on OUD outcomes among racial/ethnic minoritized groups, including examining the impact of service disruptions on vulnerable groups with OUD.

## 1. Introduction

The COVID-19 pandemic has been the most impactful health emergency in the past century. Racial/ethnic minoritized groups have been disproportionately affected by the pandemic [1,2], and there have been inequities during the vaccination rollout [3]. In the US, the pandemic has also worsened the ongoing opioid epidemic, with opioid overdoses surging during the pandemic [4]. More than 98,000 overdose deaths were reported in the first 12 months of the pandemic, with more than 70% involving opioids [5]. Racial/ethnic minoritized groups have also been increasingly affected by opioid use disorders (OUDs) [6,7], but whether these trends have persisted during the pandemic is unclear.

Measures to mitigate the COVID-19 pandemic, especially during the first months of the pandemic, have included the closure of non-essential businesses, restructuring of healthcare operations, and enactment of stay-at-home orders [8]. Some of these measures, while necessary to mitigate the pandemic, may increase the risk of negative health outcomes associated with opioid use disorders. For example, job losses and the resulting economic crisis might have led to increases in OUD cases, as previous research has linked unemployment and economic insecurity to opioid overdose deaths [9,10]. In addition, interruptions in healthcare services may increase barriers to treatment for people with OUD. While some of these closures were lifted during the summer of 2020, some changes have persisted over time. In fact, non-emergency visits to healthcare declined sharply during 2020 [11], and this may have affected people with OUD who were isolated from their outlets of support and therefore unable to receive necessary help to combat opioid misuse.

The objective of this rapid literature review is to synthesize the evidence on changes in racial/ethnic disparities in OUD-related outcomes in urban areas during the COVID-19 pandemic in the US. We conducted a rapid literature review to gain a better understanding of the research in the intersection of these two public health crises in a timely manner. Moreover, analyzing how the pandemic affected different racial groups in outcomes other than COVID-19 itself can inform broader policies and interventions that aim to minimize the public health burden on disadvantaged populations in future public health emergencies.

## 2. Methods

We searched for articles examining OUD outcomes by race/ethnicity during the COVID-19 pandemic in urban areas of the US using keywords in a structured search in PubMed. We used the following combinations of search terms:Combination 1: ((COVID-19) OR (pandemic)) AND ((opioid) OR (“overdose mortality”) OR (“SUD treatment”) OR (buprenorphine) OR (overdose)) AND ((urban) OR (cities) OR (city)).Combinations 2: ((inequity*) OR (disparity*) OR (inequality*)) AND ((COVID-19) OR (pandemic)) AND ((opioid) OR (“overdose mortality”) OR (“SUD treatment”) OR (buprenorphine) OR (overdose)) AND ((urban) OR (cities) OR (city)).

Note: We used the keyword SUD (substance use disorder) to be comprehensive and capture as much as we could in the first screening in case there were papers looking at other types of substance use disorders in addition to OUDs.

The search was conducted in PubMed on August 2021. Combinations 1 and 2 captured 71 and 5 papers, respectively, and, after removing duplicates, 71 papers remained for screening. All papers recovered using the second combination of terms (n = 5) were also recovered in the first combination. The motivation for the second search was to use the terms “inequity”, “disparity”, and “inequality” in case papers used these specific terms to characterize differences between race/ethnic groups.

We screened the papers using title and abstract to select papers for full review and extraction of results. Inclusion criteria included: (a) papers that examined racial/ethnic disparities in (b) any OUD-related outcome (e.g., opioid poisoning hospitalization or mortality or any treatment outcome, such as buprenorphine utilization) in (c) urban areas of the US, and (d) during the COVID-19 pandemic. We limited the search to urban areas because those were the areas first affected by the pandemic and where much of OUD service delivery happens. In addition, urban areas have a larger share of minoritized populations and represent highly unequal places, while at the same time having policy-making capacity [12].

Two authors, KMM and PHM, screened all articles independently and applied the inclusion/exclusion criteria. After that, the authors met to compare their results and resolved any disagreements about either inclusion or exclusion by reaching a consensus after reviewing the criteria. Both authors also extracted data from the final articles selected and met to discuss/reach consensus about any disagreements. The data extraction collected information on race/ethnicity groups included/examined, outcome(s) assessed, underlying population, e.g., geographic location or hospital catchment area, and main results related to changes in racial/ethnic inequities in OUD-related outcomes.

We reported both the race and ethnicity and the outcomes as originally reported by the articles.

## 3. Results

In total, we found 71 unique articles that were screened using the title and abstract. Of these, 54 were excluded (e.g., not US-based, not examining any outcome of interest, and those not containing data such as commentaries and protocol descriptions). A total of 17 articles were selected for full-text review, and of these, 8 articles were excluded because they did not include data stratified by race/ethnicity. Nine articles were included in the final analysis. Figure 1 summarizes the process of the inclusion/exclusion of papers.

Table 1 shows the overall characteristics of the nine articles included in this review. Of the nine studies, two were published in 2020 and seven were published in 2021. Regarding their scope, three studies examined the overall impact of the pandemic on OUD disparities [13,14,15], and six were focused on some policy or program adopted to mitigate the effect of the pandemic (four studies examined stay-at-home/shelter-in-place orders and two examined telehealth programs) [16,17,18,19,20,21]. Among the three studies examining the overall impact of the pandemic, one compared data from the first four months of the pandemic with the same period in 2019 [13], and two studies compared data from 2020 to baseline data in 2018 and 2019 [14,15]. Among the four studies examining the impact of stay-at-home orders, three included daily or monthly data from before and after the enactment of the stay-at-home order [18,19,21], and one of the studies examining the telehealth programs examined data from before and after March 2020 [16].

Regarding the racial/ethnic groups included, the first aspect to notice is that articles used inconsistent reporting of race and ethnicity. Using the original race/ethnicity reported in the article, we found that seven articles included information about white individuals [17,20], all nine articles included information about Blacks/African-American individuals, six articles contained information about Latinxs/Hispanic individuals [17,19,20], two articles included American Indian/Alaska Native individuals [13,14], three included information about Asian individuals [13,14,21], one included information about Native Hawaiian/Pacific Islander individuals [14], and five articles included other categories that differed among the sources, but included non-Black, unknown race, and multiple races [13,17,19,20,21]. Regarding the outcomes examined, three studies examined unintentional overdose deaths, either in isolation [19,21] or in combination with non-fatal overdoses [18], one study examined non-fatal overdoses in isolation [13], one study examined 911 calls and emergency medical services (EMS) activations related to opioid use [14], one examined overdose-associated cardiac arrests [15], one examined potentially harmful behavior as the result of COVID-19 mitigation policies [20], and two examined the potential of telemedicine to improve access to treatment [16,17]. Because papers reported on a variety of outcomes, we used the term “OUD-related outcome” to refer to the larger group of outcomes studied. Due to the small number of studies, we decided to include all outcomes, even though they come from different data sources and represent different burdens for the population.

Regarding the study setting, two studies used national data from the National Emergency Medical Services (EMS) Information system [14,15], and four studies used city-specific data from San Francisco [17,21], Indianapolis [19], and Philadelphia [18]. Two studies included populations served by urban healthcare facilities—a hospital [13], and an office-based opioid treatment (OBOT) program [16]—and one study examined current or former people who inject the drug (PWID) [20].

Table 2 shows the summary of the results. Six studies found an increase in opioid-related outcomes as a result of the pandemic [13,14,15] or as a result of stay-at-home orders [18,19,21]. Two studies found a reduction in barriers to treatment as a result of telehealth programs [16,17], and one study used cross-sectional data from the period during the pandemic to examine harmful behavior as the result of COVID-19 mitigation policies [20].

Regarding racial/ethnic disparities, among the six studies that found increases in OUD-related outcomes, four showed a widening of disparities in OUD-related outcomes during the pandemic due to worsening outcomes among racial/ethnic minorities [13,14,15,18], and two showed no change in distribution of outcomes by race/ethnicity before and after the pandemic [19,21]. Ochalek et al. (2020) found that, in the period between March–June 2020, 80% of patients who presented with a nonfatal opioid overdose in an urban emergency department in Virginia were Black (vs. 63% in the same period of 2019) [13]. Handberry et al. (2021) found that opioid use-related EMS activations increased the most among Asian (44.4%), Native Hawaiian/Pacific Islander (43.4%), and American Indian/Alaskan Native (41.2%) individuals, and the least among white (34.5%) and Black/African American (33.3%) individuals in 2020 compared to 2018–2019 [14]. Khatri et al. (2021) found that a stay-at-home order led to an increase in opioid overdoses among non-Hispanic Black individuals, but decreases among non-Hispanic white individuals in Philadelphia [18]. Finally, Friedman et al. (2021) found that Latinx and Black individuals had the highest percent increase in overdose-associated cardiac arrest in 2020 compared to previous years using national data [15].

Regarding the two studies examining telehealth as a way to mitigate barriers to access, O’Gurek and David (2021) found that no-show rates in an OBOT program were lower for all racial/ethnic groups after the implementation of telemedicine protocol [16], and Mehtani et al. (2021) found that, among the patients who were newly prescribed buprenorphine via telehealth, 67% were Black [17]. Finally, Genberg et al. (2021), using cross-sectional data from persons who inject drugs (PWID), found that Black PWID were more likely than non-Black to socially distance (73% vs. 48%) and inject drugs when alone (68% vs. 35%).

## 4. Discussion

In this rapid literature review examining changes in racial/ethnic disparities in OUD-related outcomes during the COVID-19 pandemic, we found general support for the hypothesis that racial/ethnic disparities have widened during this time. However, we also found that, in different geographic areas, the specific racial and ethnic minority group experiencing the worst outcomes differed. In addition, we identified a small number of studies that used before-and-after comparisons to examine the impact of policies adopted during the pandemic on disparities. Specifically, we found that stay-at-home orders have contributed to widening disparities in opioid-related outcomes. Concurrently, studies examining the use of telehealth and access to OUD treatment showed that minoritized groups have benefited from telehealth programs.

The studies we reviewed examined a variety of OUD-related outcomes, from rates of substance use to fatal opioid overdoses, and access to medication for opioid use disorder. However, only two studies used national data, and both of them found that disparities widened during the pandemic. Specifically, the study by Handberry et al. [14] found that opioid use-related activations increased more among Asian, Native Hawaiian/Pacific Islander, and American Indian/Alaskan Native individuals (compared to white individuals), and overdose-associated cardiac arrest increased the most among Latinx and Black individuals.

At the same time, local studies are key to determining the needs of local populations; for example, a Philadelphia study found that, for the first time in recent history, the absolute number of overdose deaths was higher among non-Hispanic Black individuals than among non-Hispanic white individuals [18]. However, studies that include a large number of cities or jurisdictions can be used to characterize heterogeneity across geographic areas, and point to potentially successful initiatives in specific jurisdictions. The availability of timely data is an important limitation to conducting these studies, and a coordinated effort is needed to produce such data, particularly in the context of these two health emergencies.

The combination of the opioid epidemic and the COVID-19 pandemic could be conceptualized as a syndemic, which is characterized by biological and social interactions between conditions that increase people’s susceptibility to harm or worse health outcomes [22]. People with OUD are at a higher risk of contracting and having more severe COVID-19, with a particularly higher risk for Black individuals [23], given the centuries of structural racism that have generated unequal living conditions for the African American population [24]. In general, people with an OUD diagnosis are 2.4 times more likely to have had a COVID-19 diagnosis than those without, and patients with lifetime substance use disorders experience more severe outcomes [23]. Underlying pathways that explain this worsening of health among people with OUD include biological and social pathways, related to the impact of opioid use on the respiratory system [25], stigma [26], and a higher risk of exposure among already vulnerable populations, such as those economically disadvantaged and racial/ethnic minorities [27]. This is consistent with the finding by Genberg et al. [20], which indicates that Black PWID were more likely than non-Black to socially distance, maybe due to a higher awareness of COVID-19, and to inject drugs when alone. According to the fundamental causes theory [28], this combination of socially driven exposures puts minoritized groups at a higher risk for an extensive number of health outcomes, as the same systems of exposure [29] giving rise to increased opioid use may also give rise to an increased exposure to COVID-19. For example, service workers are at an increased risk for both drug overdose mortality [30] and COVID-19 [31], and are disproportionately likely to be minoritized [31].

Given this context, policies to address the pandemic need to consider the potential negative impacts of public health interventions, such as stay-at-home orders on people with OUD. In early 2021, approximately 4 in 10 adults in the US reported symptoms related to anxiety and depression, compared to only 1 in 10 in early 2019 [32]. This increase has been linked to worry or stress related to the risk of infection, as well as to other factors, such as job loss and isolation [33]. In the context of people with OUD, these factors increase the risk of overdose. However, more rigorous research is needed to understand how stay-at-home orders and closures affected or interacted with the closure of outpatient locations where people with OUD receive treatment [34].

Telemedicine has been increasingly described as a solution for improving access to OUD treatment, and there is a growing body of research on the use of telemedicine during the pandemic and beyond [35]. Treatment initiation via telemedicine was found to increase access to treatment from home using buprenorphine [36]. However, more research is needed to understand whether there is variation in the use of telemedicine across race/ethnicity groups that could result in disparities. In addition, it is important to keep in mind that the use of telemedicine to treat OUD was allowed due to the temporary relaxation of federal and state regulations [37] under the public health emergency declaration. In that sense, the future of telehealth use in OUD treatment is uncertain once the public health emergency declaration is lifted. This may also reduce Medicaid coverage [38], resulting in gaps in access to care, especially for low-income and minoritized populations. In addition to telehealth, policies to mitigate opioid harm, such as supervised injection sites [39] and expanding the treatment/removal of barriers [40], could reduce the risk of overdoses among this population.

The studies we compiled and our review have several limitations. First, given the heterogeneity in studies, especially in their design and scope, assessing their risk of bias is challenging. Second, the studies were not able to or did not aim to separate drug overdoses by drug type, not allowing us to examine opioid overdoses specifically. However, according to the CDC, opioid overdoses represent more than 70% of all drug overdoses in the US [5]. Due to the small number of studies that we were able to find, we decided to include all of these outcomes, even though they may represent different proportions of opioid overdoses. Third, the reporting of race/ethnicity was heterogeneous, with studies using different groups, making comparisons across them unfeasible. We also excluded a number of articles examining the intersection between the COVID-19 pandemic and the opioid epidemic because they did not present data stratified by race/ethnicity at all. Even in the final set of studies, only a small set was focused on assessing disparities or changes in disparities over time, and not all studies included data from before the pandemic as a baseline for comparison. This finding is consistent with another review examining the impact of the pandemic in OUD treatment delivery, which found that, among the 14 studies included, only 2 (which were also included in our review) examined differences by race/ethnicity [36]. The small number of studies is a limitation of this review, but it is also an indication that there is much to be explored in the intersection between the opioid epidemic and disparities. Disparities in COVID-19 have been a focal point of a growing number of studies [27,41]. However, less attention has been paid to the way in which COVID-19 disparities can affect other population health disparities, including among people with OUD. Last, we restricted our search to the peer-reviewed literature and did not examine guidelines put forward by state or local health departments. Public health departments have put guidelines in place to reduce stigma related to OUD treatment, as well as equity-based guidelines and policies to mitigate the uneven burden of disease among minoritized groups [42]. However, research is needed to understand the extent to which these guidelines are translated into the actual treatment of minoritized groups from an equity perspective.

## 5. Contribution

Our study demonstrates a widening of the racial/ethnic gap in OUD-related outcomes due to worsening outcomes among Black, Asian, and Latinx individuals during the pandemic. This finding contributes to a larger body of evidence on racial disparities in health and raises awareness of OUD among racial/ethnic minoritized groups, an issue that has been associated with whites in rural areas of the country for more than two decades.

## 6. Future Research Directions

More research is needed to understand the impact of the COVID-19 pandemic and related policies on OUD outcomes among racial/ethnic minoritized groups. During the pandemic, a number of policies were adopted to reduce barriers to treatment for opioid use disorders. Such policies included the removal of the requirement for an in-person visit before starting buprenorphine and allowing for methadone treatment using telehealth [43]. Examining the effect of these policies across racial/ethnic minoritized groups can inform efforts to reduce disparities and improve health among people with opioid use disorders. The use of causal methods for these evaluations can enhance their usefulness [44], but care must be taken when evaluating policies that co-occur with other policies (e.g., shelter at home concurrent with closures of certain outpatient locations) [45]. In addition, researchers should aim for consistency in reporting race and ethnicity in scientific manuscripts [46].

## 7. Policy Recommendations

Policy makers should plan to attend to the needs of vulnerable populations with OUD. These needs range from addressing some of the social determinants of health, such as improvements in living conditions in urban neighborhoods, to offering medication-based treatment to those with OUD. Policy makers should also carefully balance the potential positive and negative effects of public health interventions in the face of public health emergencies. They should aim to achieve a better balance between the risk for the general population versus the risk for groups that are especially vulnerable, such as people with OUD, by supporting the continuation and strengthening of vital services in urban communities. The lack of such support measures has been linked to the recent increase in deaths of despair, mostly fueled by opioid overdoses [47].

## 8. Conclusions

In summary, we found evidence that generally demonstrates a widening of the racial/ethnic gap in OUD-related outcomes during the pandemic. However, the limited number of studies in a small number of jurisdictions indicates a gap in research examining the intersection between COVID-19 and OUD-related outcomes with a focus on disparities. Policies to address the pandemic need to consider the potential negative effects of public health interventions, such as stay-at-home orders, in the context of people with OUD.

## Figures and Tables

**Figure 1 ijerph-19-09283-f001:**
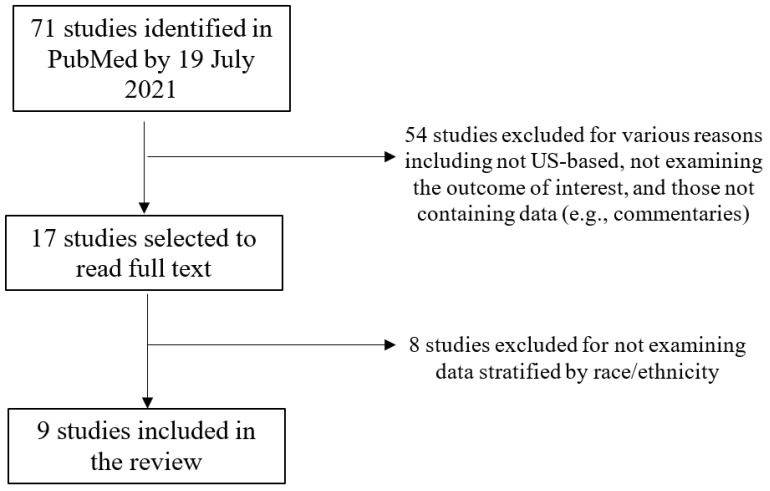
Flowchart of study inclusion.

**Table 1 ijerph-19-09283-t001:** Overall characteristics and scope of the studies.

Study	Scope	Period	Race/Ethnicity Groups	Outcomes	Study Setting/Data Source
Friedman et al., 2021 [15]	Overall impact of the pandemic.	Data from 2020 were compared to baseline values in 2018 and 2019.	Latinx, Black, White	Overdose-associated cardiac arrests	US population. Data were from the National Emergency Medical Services Information system.
Handberry et al., 2021 [14]	Overall impact of the pandemic.	Data from 2020 were compared to baseline values in 2018 and 2019.	American Indian/Alaska Native, Asian, Black/African American, Hispanic/Latino, Native Hawaiian/Pacific Islander, and White.	911 calls/EMS activations related to opioid use	US population. Data were from the National Emergency Medical Services Information system.
Ochalek et al., 2020 [13]	Overall impact of the pandemic.	Data from 1 March to 30 June 2020, were compared to the same period in 2019.	Black/African American, White, Hispanic, Asian, American Indian or Alaska Native, more than 1 race/ethnicity	Non-fatal opioid overdoses	Population served by an urban emergency department in Virginia. Data were from electronic medical records.
Appa et al., 2021 [21]	Specific to mitigation policies (shelter-in-place order).	2019–2020 (monthly data). Specifically, 8.5 calendar months before and after the shelter-in-place order on 17 March 2020.	Asian, Black, White/Latinx, Other, Unknown	Unintentional overdose deaths	San Francisco population. Data were from the Office of the Chief Medical Examiner.
Glober et al., 2020 [19]	Specific to mitigation policies (stay-at-home order).	2019–July 2020 (daily data). Data were analyzed in two periods: 122-day period directly preceding the stay-at-home order on 25 March, 2020, and the 122-day period after the order.	Black, White, Other	Unintentional overdose deaths	Indianapolis population. Data was from EMS System.
Genberg et al., 2021 [20]	Specific to mitigation policies (stay-at-home).	April–June 2020	Black, Non-Black	Substance use rates	Current and former people who inject drugs in Baltimore. Data were from a phone survey.
Khatri et al., 2021 [18]	Specific to mitigation policies (stay-at-home).	2019–June 2020 (monthly data). Data were analyzed in three periods, before and after the stay-at-home in March 2020, and April to June 2019.	Non-Hispanic Black, Non-Hispanic White, Hispanic	Unintentional overdose deaths and nonfatal overdoses	Philadelphia population. Data on overdose deaths were from the Philadelphia Medical Examiner’s Office and non-fatal overdoses were from EMS.
Mehtani et al., 2021 [17]	Specific to mitigation policies (Telehealth).	10 April–25 May 2020.	Black and Non-Black	Untreated OUD in isolation and quarantine sites	San Francisco population. Data were from telehealth encounters from electronic medical record and data from the San Francisco Department of Public Health.
O’Gurek and David, 2021 [16]	Specific to mitigation policies (Telehealth).	Data from 16 March–30 April 2020, were compared to data from 1 January–13 March 2020.	White, Hispanic non-white, African American	Medical center visits	Population served by an urban office-based opioid treatment program. Data were from patient chart review.

Notes: EMS—emergency medical service.

**Table 2 ijerph-19-09283-t002:** Summary of study results.

Study	Results Summary	Result (More Detail)
Ochalek et al., 2020 [13]	Overall: Increase in opioid-related outcomes during the pandemic. Disparity specific: Outcomes more common among Black patients vs. white (compared to before).	The total number of nonfatal opioid overdose visits increased during the pandemic. Among patients who presented with a nonfatal opioid overdose in 2019 and 2020, 64 (63%) and 181 (80%) were Black, respectively, vs. 29 (28%) and 32 (14%) who were white.
Handberry et al., 2021 [14]	Overall: Increase in opioid-related outcomes during the pandemic. Disparity specific: Larger increases among Asian, Pacific Islander, and American Indian/Alaskan Native.	911 opioid use/overdose EMS activations were higher starting on March 2020 compared to pre-pandemic levels. When examined by race/ethnicity, opioid-use-related activations increased more among Asian, Native Hawaiian/Pacific Islander, and Alaskan Indian/American Native, and least among white and Black/African American.
Khatri et al., 2021 [18]	Overall: Increase in opioid-related outcomes after stay-at-home order. Disparity specific: Outcomes more common among Black patients vs. white (compared to before).	The stay-at-home order was associated with increases in opioid overdose among non-Hispanic Black individuals but decreases among non-Hispanic white individuals. This represents the first time in recent history in Philadelphia that the absolute number of overdose deaths was higher among non-Hispanic Black individuals than among non-Hispanic white individuals.
Friedman et al., 2021 [15]	Overall: Increase in opioid-related outcomes during the pandemic. Disparity specific: Outcome increased more among Latinx and Black individuals	Overdose-associated cardiac arrests increased by 42.1% nationally in 2020. The highest percentage increases were seen among Latinx and Black individuals.
Appa et al., 2021 [21]	Overall: Increase in opioid-related outcomes after shelter-in-place order. Disparity specific: No change in distribution by race/ethnicity.	The median number of weekly overdose deaths increased by 50% after the shelter-in-place order. While the proportion of Black decedents slightly decreased after the shelter-in-place order, the death rate was still disproportionately high for Black vs. white residents.
Glober et al., 2020 [19]	Overall: Increase in opioid-related outcomes after stay-at-home order. Disparity specific: No change in distribution by race/ethnicity.	Overdose-related calls for EMS and EMS naloxone administration increased during and after the Indiana stay-at-home order. Overdose-related calls increased by 43% and naloxone administration increased by 61% after the stay-at-home order. Deaths from drug overdoses increased by 47%. There was no change in distribution of age, race/ethnicity, or zip code of those who overdosed after the stay-at-home order was issued.
O’Gurek and David, 2021 [16]	Overall: Decline in opioid-related outcomes after implementation of a telemedicine protocol. Disparity specific: Declines were seen for all race/ethnic groups.	No-show rate in the office-based opioid treatment program reduced from 26% to 8% after the implementation of a telemedicine protocol. No-show rates reduced for all race/ethnicity groups.
Mehtani et al., 2021 [17]	Overall: Decline in opioid-related outcomes after implementation of telehealth program. Disparity specific: Vulnerable population targeted by the program were mostly Black.	Telehealth was used on the management substance use disorders for 59 guests in isolation and quarantine sites. Twelve patients were identified with untreated opioid use disorder and newly prescribed buprenorphine. Of these, all were marginally housed, 67% were Black, and 58% had never previously been prescribed medications for OUD.
Genberg et al., 2021 [20]	Overall: Current and former PWID reporting substance use were less likely to always social distance. Disparity specific: Black PWID were more likely than non-Black to socially distance and use when alone.	Black individuals who inject drugs were more likely than non-Black individuals to socially distance (73% vs. 48%) and use when alone (68% vs.35%).

Notes: EMS—emergency medical service; PWID—people who inject drugs.

## Data Availability

No new data were created or analyzed in this study besides the information presented in the tables. Data sharing is not applicable to this article.

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
