# Peer review of "Changes in Racial/Ethnic Disparities in Opioid-Related Outcomes in Urban Areas during the COVID-19 Pandemic: A Rapid Review of the Literature"

_ijerph, 2022, doi:10.3390/ijerph19159283_

Round 1

Reviewer 1 Report

Review Report

Moran et al present a review on “Changes in racial/ethnic disparities in opioid-related outcomes in urban areas during the COVID-19 pandemic”. The manuscript is well written. There are, however, a few revisions that the authors need to make, following which the manuscript may be accepted for publication. These are as follows:

Line 53: Kindly consider replacing “the COVID-19 pandemic” with “it”.

Line 55: Kindly consider replacing “the COVID-19 pandemic” with “the pandemic”.

Line 62: Please rewrite “might led to increases in OUD” as “might lead to an increase in OUD cases”.

Line 63: Please change “has” to “have”.

Line 198: Please rewrite “In additional” as “In addition”.

Line 207: Please rewrite “Handberry et al. (2021)12” as “Handberry et al. 12”.

Line 224: Please delete the first comma in “individuals,21 , given” as well as the space preceding the second comma.

Line 231: Please rewrite “population” as “populations,”.

Lines 232–233: Please rewrite “by Gengberg et al. (2021)18” as “of Gengberg et al. 18” and “non-Black” as “non-black PWID”.

Line 236: Please rewrite “impact” as “impacts” and “measures such as” as “public health interventions like”.

Lines 238–241: Please precede each “such as” with a comma.

Author Response

Reviewer 1:

Line 53: Kindly consider replacing “the COVID-19 pandemic” with “it”.

Line 55: Kindly consider replacing “the COVID-19 pandemic” with “the pandemic”.

Line 62: Please rewrite “might led to increases in OUD” as “might lead to an increase in OUD cases”.

Line 63: Please change “has” to “have”.

Line 198: Please rewrite “In additional” as “In addition”.

Line 207: Please rewrite “Handberry et al. (2021)12” as “Handberry et al. 12”.

Line 224: Please delete the first comma in “individuals,21 , given” as well as the space preceding the second comma.

Line 231: Please rewrite “population” as “populations,”.

R: We have fixed all these issues identified by the reviewer.

Lines 232–233: Please rewrite “by Gengberg et al. (2021)18” as “of Gengberg et al. 18” and “non-Black” as “non-black PWID”.

R: We have left “Black” capitalized, consistent with the some commonly used guidelines:

https://apnews.com/article/archive-race-and-ethnicity-9105661462

https://www.nytimes.com/2020/07/05/insider/capitalized-black.html

Line 236: Please rewrite “impact” as “impacts” and “measures such as” as “public health interventions like”.

Lines 238–241: Please precede each “such as” with a comma.

R: We have fixed both issues.

Please see revised document attached.

Reviewer 2 Report

I find the topic of the reviewed review, entitled Changes in racial/ethnic disparities in opioid-related outcomes in urban areas during the COVID-19 pandemic: a rapid review of the literature, interesting and up-to-date. The review is based on only nine research studies selected from the group of 71 studies. The structure of the entire manuscript is adequate. However, its quality could be improved.

Below there is a list of my critical remarks on the reviewed manuscript (review):

·        Please rearrange the abstract to make it shorter. Do not distinguish parts of the abstract.

·        The introduction section should/could be developed.

·        Please present the same precise aim in the text and the abstract.

·        Please present the contribution of the study.

·        The discussion section should include the discussion with other studies (use them as references). Please elaborate on this issue.

·        Please formulate (short) recommendations for decision-makers (based on your results).

Overall assessment

I find the reviewed study valuable and interesting, and its topic is up-to-date. The quality of this review is average. I recommend making changes (considering the remarks mentioned above) that should/could improve the quality of the reviewed manuscript.

Author Response

Reviewer 2:

I find the topic of the reviewed review, entitled Changes in racial/ethnic disparities in opioid-related outcomes in urban areas during the COVID-19 pandemic: a rapid review of the literature, interesting and up-to-date. The review is based on only nine research studies selected from the group of 71 studies. The structure of the entire manuscript is adequate. However, its quality could be improved.

Below there is a list of my critical remarks on the reviewed manuscript (review):

  • Please rearrange the abstract to make it shorter. Do not distinguish parts of the abstract.

R: We have rewritten the abstract to keep it below 200 words:

Abstract: Opioid use disorders (OUDs) are increasingly more common among minoritized populations, who have historically experienced limited access to healthcare, a situation that may have worsened during the COVID-19 pandemic. Using a structured keyword search in Pubmed, we reviewed the literature to synthesize the evidence on changes in racial/ethnic disparities in OUD-related outcomes in urban areas during the COVID-19 pandemic in the US.  Nine articles were included in the final analysis. Six found increases in OUD-related outcomes during the pandemic, with four showing a widening of disparities. Results also point to worsening of opioid outcomes among Black and Latinx individuals related to shelter-in-place or stay-at-home orders. Studies examining the use of telehealth and access to OUD treatment showed that minoritized groups have benefited from telehealth programs. The limited number of studies and the inclusion of only a few jurisdictions indicates a gap in research examining the intersection between COVID-19 and OUD-related outcomes with a focus on disparities. More research is needed to understand the impact of the COVID-19 pandemic and related policies on OUD outcomes among racial/ethnic minoritized groups, including examining the impact of service disruptions on vulnerable groups with OUD.

  • The introduction section should/could be developed.

R: We have slightly restructured a few parts of the introduction to make it more clear.

  • Please present the same precise aim in the text and the abstract.

R: We have now done that.

  • Please present the contribution of the study.

R: We have now included a section “contribution”, lines 333-338.

  • The discussion section should include the discussion with other studies (use them as references). Please elaborate on this issue.

R: We have now expanded the discussion.

  • Please formulate (short) recommendations for decision-makers (based on your results).

R: We have now included a section “Policy recommendations”, lines 351-361.

Overall assessment

I find the reviewed study valuable and interesting, and its topic is up-to-date. The quality of this review is average. I recommend making changes (considering the remarks mentioned above) that should/could improve the quality of the reviewed manuscript.

R: We thank the reviewer for these suggestions.

Reviewer 3 Report

Historically, opioid use disorder (OUD) and OUD-associated health disparities are more common among minoritized racial/ethnic groups. More recently, these groups have been disproportionally affected by the COVID-19 pandemic. In the United States, there is evidence to suggest that the COVID-19 pandemic worsened the ongoing opioid epidemic, which poses a threat to public health, especially amongst minoritized racial/ethnic groups. To further explore this topic, researchers undertook a rapid review of the literature to examine changes in racial/ethnic disparities in OUD-related outcomes in the context of the COVID-19 pandemic, with a focus on urban areas in the United States.

The reviewer offers the following comments, by section of the manuscript, for the authors’ consideration.

Abstract

In the third sentence of the Introduction, consider changing “epidemic” to “pandemic” in reference to the COVID-19 global crisis (line 20).

In general, the Conclusion is very lengthy. The authors might consider shortening the Conclusion by focusing on the most salient (one to three) points that summarize their research findings and implications. For example, points about more research or future research might be more appropriate for the manuscript Discussion than the abstract Conclusion.

Methods 

As a point of clarification, did the Combination 2 search yield only 5 results in total or 5 results that were distinct from the Combination 1 search (i.e., 71 original results [Combination 1] + 5 distinct results [Combination 2] = 76 results total [Combination 2])? Perhaps the addition of the Boolean Operator “AND” before the COVID-19 search term significantly narrowed down the search results for Combination 2. However, this might warrant clarification for the readership, or at least an explanation of the rationale for the different searches.

Results

The last sentence of the first paragraph of the Results section (page 3, lines 111-115) is essentially a repeat of the last sentence of the Methods section (page 3, 101-104). The authors should consider omitting this [repetitive] information from the Results, considering that this information is better suited for the Methods.

While the reviewer appreciates the listing of published studies (presumably) in chronological order within Table 2 (page 4), this table might be improved by organizing study data according to their scope. For example, the authors could consider grouping the three studies that examined the overall impact of the COVID-19 pandemic on OUD disparities together but separated from the six studies that focused on some policy or program adopted to mitigate the effect of the pandemic.

In general, it is worth pointing out that, according to the study titles (Table 2) and the study outcomes (Table 3) alone, several of the included studies (e.g., references 13, 17, and 19) appear to have a broad focus on drug overdoses and not specifically opioid-related drug overdoses. Given that this rapid literature review aimed to assess changes in racial/ethnic disparities in opioid-related outcomes during the COVID-19 pandemic, it would behoove the authors to clarify whether and to what extent (if applicable) these studies included people who suffered from non-opioid-related outcomes (e.g., alcohol- and/or benzodiazepine-related overdoses). Inclusion of such populations could potentially confound the results of this review.

Discussion

The authors point out that people with OUD are at a higher risk of contracting and having more severe COVID-19, with particularly higher risk for African Americans/Blacks (pages 7-8, lines 223-228). Further, they mention a few underlying biological and social pathways that explain this worsening of health among people with OUD (page 8, lines 228-232). Given the importance of the noted associations between OUD diagnosis and COVID-19 diagnosis, especially amongst racially/ethnically disparate groups such as African Americans/Blacks, it would seem prudent that the authors elaborate on the explanation. In other words, the Discussion section would benefit from a deeper dive into associated and/or causative factors between living with OUD, contracting COVID-19 infection, and experiencing more severe COVID-19, particularly from a social determinants of health perspective.

While the researchers are commended for identifying several limitations of their review, this section of the Discussion (page 8, lines 244-251) could be improved by (a) providing insight into the implications of their study limitations and (b) critiquing some of the limitations of the nine studies included in this review. Regarding the latter, the researchers might consider addressing some of the well-recognized types of bias that commonly occur with retrospective, cohort, and/or cross-sectional studies (e.g., selection bias, immortal time bias), if applicable, and moreover how such bias may have affected study findings and/or this review.

Author Response

Reviewer 3:

Historically, opioid use disorder (OUD) and OUD-associated health disparities are more common among minoritized racial/ethnic groups. More recently, these groups have been disproportionally affected by the COVID-19 pandemic. In the United States, there is evidence to suggest that the COVID-19 pandemic worsened the ongoing opioid epidemic, which poses a threat to public health, especially amongst minoritized racial/ethnic groups. To further explore this topic, researchers undertook a rapid review of the literature to examine changes in racial/ethnic disparities in OUD-related outcomes in the context of the COVID-19 pandemic, with a focus on urban areas in the United States.

The reviewer offers the following comments, by section of the manuscript, for the authors’ consideration.

R: We thank the reviewer for these suggestions.

Abstract

In the third sentence of the Introduction, consider changing “epidemic” to “pandemic” in reference to the COVID-19 global crisis (line 20).

R: We have now done that.

In general, the Conclusion is very lengthy. The authors might consider shortening the Conclusion by focusing on the most salient (one to three) points that summarize their research findings and implications. For example, points about more research or future research might be more appropriate for the manuscript Discussion than the abstract Conclusion.

R: We have now shortened the abstract.

Methods 

As a point of clarification, did the Combination 2 search yield only 5 results in total or 5 results that were distinct from the Combination 1 search (i.e., 71 original results [Combination 1] + 5 distinct results [Combination 2] = 76 results total [Combination 2])? Perhaps the addition of the Boolean Operator “AND” before the COVID-19 search term significantly narrowed down the search results for Combination 2. However, this might warrant clarification for the readership, or at least an explanation of the rationale for the different searches.

R: All papers recovered in the second search were also recovered in the first search. The motivation for the second search was to use the terms “inequity”, “disparity” and inequality”, in case papers used these specific terms to characterize differences between race/ethnic groups. We have added this explanation to the methods section (line 107-110)

Results

The last sentence of the first paragraph of the Results section (page 3, lines 111-115) is essentially a repeat of the last sentence of the Methods section (page 3, 101-104). The authors should consider omitting this [repetitive] information from the Results, considering that this information is better suited for the Methods.

R: We have now consolidated this information in the methods section and deleted it from the results (see marks line 13-141).

While the reviewer appreciates the listing of published studies (presumably) in chronological order within Table 2 (page 4), this table might be improved by organizing study data according to their scope. For example, the authors could consider grouping the three studies that examined the overall impact of the COVID-19 pandemic on OUD disparities together but separated from the six studies that focused on some policy or program adopted to mitigate the effect of the pandemic.

R: We have now grouped the studies by scope as recommended by the reviewer (new table 2).

In general, it is worth pointing out that, according to the study titles (Table 2) and the study outcomes (Table 3) alone, several of the included studies (e.g., references 13, 17, and 19) appear to have a broad focus on drug overdoses and not specifically opioid-related drug overdoses. Given that this rapid literature review aimed to assess changes in racial/ethnic disparities in opioid-related outcomes during the COVID-19 pandemic, it would behoove the authors to clarify whether and to what extent (if applicable) these studies included people who suffered from non-opioid-related outcomes (e.g., alcohol- and/or benzodiazepine-related overdoses). Inclusion of such populations could potentially confound the results of this review.

R: The reviewer is correct that we do not know the share of opioid-related outcomes. We have highlighted that in our limitations (lines 305-309). This is a limitation in many studies focusing on specific types of drug overdoses because there are challenges in identifying the specific drug involved and also because many times different drugs could be involved. However, opioid represent more than 70% of all drug overdoses in the US. Due to the small number of studies, we decided for including all these outcomes even though they may represent different proportions of opioid overdoses.

Discussion

The authors point out that people with OUD are at a higher risk of contracting and having more severe COVID-19, with particularly higher risk for African Americans/Blacks (pages 7-8, lines 223-228). Further, they mention a few underlying biological and social pathways that explain this worsening of health among people with OUD (page 8, lines 228-232). Given the importance of the noted associations between OUD diagnosis and COVID-19 diagnosis, especially amongst racially/ethnically disparate groups such as African Americans/Blacks, it would seem prudent that the authors elaborate on the explanation. In other words, the Discussion section would benefit from a deeper dive into associated and/or causative factors between living with OUD, contracting COVID-19 infection, and experiencing more severe COVID-19, particularly from a social determinants of health perspective.

R: We have added a few sentences in the discussion explaining more about this mechanism (lines 266-271).

While the researchers are commended for identifying several limitations of their review, this section of the Discussion (page 8, lines 244-251) could be improved by (a) providing insight into the implications of their study limitations and (b) critiquing some of the limitations of the nine studies included in this review. Regarding the latter, the researchers might consider addressing some of the well-recognized types of bias that commonly occur with retrospective, cohort, and/or cross-sectional studies (e.g., selection bias, immortal time bias), if applicable, and moreover how such bias may have affected study findings and/or this review.

R: We have now expanded our limitation section (lines 303-326).

Please see attached revised manuscript with track changes

Reviewer 4 Report

This manuscript has presented a review of changes in racial/ethnic disparities in opioid-related outcomes in urban areas during the COVID-19 pandemic. The topic is relevant but the manuscript needs to be worked on.  The abstract is too long, it needs to be simplified

In general, all the manuscript requires extensive editing of the English language and style. There are phrases started by “AND”, and this is not scientific writing

The authors have used some abbreviations that need to be spelt out, when they first appear, such as SUD (line83) and EMS (line 154).

Figure 1- does not add anything to what has already been mentioned in the text, it should therefore be deleted.

Regarding tables 2, 3 and 4, as is information for all the articles included, in my opinion, they should all be aggregated in a single table, with simplified information, and some columns may be eliminated, such as the title of the articles. On the other hand, all abbreviations must be spelt in footnotes, regardless of whether they have already been spelt in the text. The table should in the end be a concise and effective way to present large amounts of data.

Author Response

Reviewer 4:

This manuscript has presented a review of changes in racial/ethnic disparities in opioid-related outcomes in urban areas during the COVID-19 pandemic. The topic is relevant but the manuscript needs to be worked on.  The abstract is too long, it needs to be simplified

In general, all the manuscript requires extensive editing of the English language and style. There are phrases started by “AND”, and this is not scientific writing

R: We have now fixed that.

The authors have used some abbreviations that need to be spelt out, when they first appear, such as SUD (line83) and EMS (line 154).

R: We have now done that (track change lines 101 for SUD and 177 for EMS).

Figure 1- does not add anything to what has already been mentioned in the text, it should therefore be deleted.

R: We decided to keep Figure 1 even though it is a bit repetitive with the text because that is the flowchart of study inclusion and an usually required figure in literature reviews. We are open to removing it if the editor considers it unimportant.

Regarding tables 2, 3 and 4, as is information for all the articles included, in my opinion, they should all be aggregated in a single table, with simplified information, and some columns may be eliminated, such as the title of the articles. On the other hand, all abbreviations must be spelt in footnotes, regardless of whether they have already been spelt in the text. The table should in the end be a concise and effective way to present large amounts of data.

R: We have now aggregated information from tables 2 and 3 (now table 2) and eliminated the titles of the articles. We have also spelled out all the abbreviations. We have placed the tables in the body of the article following the journal’s guidelines.

We thank the reviewer for this suggestions. Please find the revised manuscript with track changes attached. 

Reviewer 5 Report

The authors submitted a "rapid" review regarding potential disparities in opioid related “outcomes” in urban US.

The topic is of high interest to the readers of IJERPH because of the ongoing opioid crisis in US and possibly in other areas all over the world.

Unfortunately, it is hard to understand authors approach.

1) Rapid communication is common for outstanding findings, under acute conditions, and in short form.

There is no “short” review. Considering the long-lived nature of the opioid crisis (and the now 2y long COVID-19 epidemic), an in-depth and comparative analysis was expected, and demanded. Overall, it is a rush job.

Opioid crisis started has been linked to prescription, and probably non-prescription, opioid drug misuse.

HHS has a special site

https://www.hhs.gov/opioids/about-the-epidemic/index.html

there is an observatory

https://www.cdc.gov/nchs/pressroom/nchs_press_releases/2021/20211117.htm

reports have been released amidst the COVID-19 epidemic, and there are regular updates from the national health services (Jan 2022):

https://nchstats.com/category/opioid/

States may also provide live data and access to data:

https://www.michigan.gov/mdhhs/safety-injury-prev/environmental-health/topics/mitracking/overdose

2) Specific protocols regarding the COVID-19 pandemic and the substance use stigma have been issued:

https://www.naccho.org/programs/community-health/injury-and-violence/opioid-epidemic

The details about “spikes” and equity policy are not mentioned, and far than integrated into the manuscript.

3) Because of the above vast list of resources and frameworks, it is hard to believe that we lack information about populations groups.

Because of the equity policy, it is also hard to understand why the authors (and literature) do not follow the standards regarding race and ethnicity

https://grants.nih.gov/grants/guide/notice-files/not-od-15-089.html

and the updated JAMA scope:

https://jamanetwork.com/journals/jama/fullarticle/2783090

4) SUD (substance use disorder) is suddenly introduced as a key word, and it is basically a general term.

Subscription, or any relevant medical history is not explored.

Outcomes is a general term. In Pharmacology it is not valid term. Could be just opioid overdose, or opioid overdose deaths, opioid overdose management or ....

The terminology is common in all reports above, and authors approach seems irrelevant to the field.

5) The last major issue is the focus in urban areas. Actually, there is none, or a justification of this type of “review”/search combinations, and future policies.

6) I wish I wouldn’t add ethical issues to the list. I am just wondering who would benefit from this work, and why official reports and comparisons have been omitted.

Author Response

Reviewer 5:

The authors submitted a "rapid" review regarding potential disparities in opioid related “outcomes” in urban US.

The topic is of high interest to the readers of IJERPH because of the ongoing opioid crisis in US and possibly in other areas all over the world.

Unfortunately, it is hard to understand authors approach.

1) Rapid communication is common for outstanding findings, under acute conditions, and in short form.

There is no “short” review. Considering the long-lived nature of the opioid crisis (and the now 2y long COVID-19 epidemic), an in-depth and comparative analysis was expected, and demanded. Overall, it is a rush job.

Opioid crisis started has been linked to prescription, and probably non-prescription, opioid drug misuse.

HHS has a special site

https://www.hhs.gov/opioids/about-the-epidemic/index.html

there is an observatory

https://www.cdc.gov/nchs/pressroom/nchs_press_releases/2021/20211117.htm

reports have been released amidst the COVID-19 epidemic, and there are regular updates from the national health services (Jan 2022):

https://nchstats.com/category/opioid/

States may also provide live data and access to data:

https://www.michigan.gov/mdhhs/safety-injury-prev/environmental-health/topics/mitracking/overdose

R: We appreciate the sources of data shared by the reviewer. We have re-examined them in search for data stratified by race/ethnicity specifically in urban areas, but have not found such data. We understand that there are original sources of data that can be used to conduct such analysis. However, our goal was not to write an original manuscript, but to conduct a literature review of what has been published on OUD-related disparities during the pandemic. Reviews of the literature have a role of not just describing a public health problem, but also to give a sense of “what is” and “what is not” the focus on the literature. In this case, we conducted a rapid literature review to create “evidence synthesis that may provide more timely information for decision making compared with standard systematic reviews” https://www.ncbi.nlm.nih.gov/books/NBK362003/#:~:text=Rapid%20reviews%20are%20a%20form,compared%20with%20standard%20systematic%20reviews.

We acknowledge that the timeframe of the review was longer than recommended due to the analysis and drafting of the paper taking longer than originally planned. However, we still think our results make a contribution to the field due to our specific and unique focus and the small number of studies in this particular intersection between OUD, disparities, and COVID-19 pandemic.

The reviewer is correct in that there is an abundance of data on the opioid epidemic. However, our review shows that, there is a lack of studies examining changes in OUD disparities that may have been related to the COVID-19 pandemic. This finding is consistent with another review examining the impact of the pandemic in OUD treatment delivery, which found that among the 14 studies included, only 2 (which were also included in our review) examined differences by race/ethnicity. We have highlighted these findings in the discussion (lines: 316-319).

Our review also summarizes the source of information and the outcomes examined during the period studies. These data sources may represent sources that were more quickly available during the crises. Many of the studies reviewed used data from local medical examiner’s offices and electronic medical records. These are not data sources that are easily available online. Data sources indicated by the reviewer, such as mortality data were not available to researchers until more recently, almost two years after the start of the pandemic. In addition, many of these data sources do not provide data stratified by race/ethnicity. This is a limitation that should be addressed.

2) Specific protocols regarding the COVID-19 pandemic and the substance use stigma have been issued:

https://www.naccho.org/programs/community-health/injury-and-violence/opioid-epidemic

The details about “spikes” and equity policy are not mentioned, and far than integrated into the manuscript.

R: Our approach was to synthesize the peer-reviewed literature. We did not aim to synthesize reports on equity policies released during the pandemic. These policies have good intention and might, in the future, lead to better health equity, but that was not the goal of this review. Provisional death counts released by the CDC are helpful, but the dashboard mentioned doesn’t always allow for the stratification of data by race/ethnicity or by geography, including cities, which was our focus.

We have added more background on the “spikes” (lines 65-68) but we believe that there is already a rich literature on that. We wanted to focus on changes in disparities in OUD potentially related to the pandemic, which we believe deserves more attention. We have also added a few sentences in the discussion mentioning the efforts towards equity policies of various agencies and organizations (lines 326-330).

3) Because of the above vast list of resources and frameworks, it is hard to believe that we lack information about populations groups.

Because of the equity policy, it is also hard to understand why the authors (and literature) do not follow the standards regarding race and ethnicity

https://grants.nih.gov/grants/guide/notice-files/not-od-15-089.html

and the updated JAMA scope:

https://jamanetwork.com/journals/jama/fullarticle/2783090

R: As previously mentioned, our goal was to synthesize the findings specifically on OUD-related outcomes during the pandemic. The reviewer is correct that there is a vast literature on health disparities, but we have identified only a small number of papers examining outcomes in the intersection between the OUD epidemic and the COVID-19 pandemic. And even a smaller share presented race/ethnicity stratified results, hence the need for such a review.

The reviewed is also correct in his remark regarding the lack of standardization on race/ethnicity groups. In fact, we believe that our review has the potential to make this issue more visible. We reported the exact terms used in the papers instead of renaming the groups based on the official recommendations (JAMA paper and NIH guidelines). We have added a few sentences in the methods section (to clarify our decision to report the race/ethnicity terms originally described in the papers – line 128), results (line 166), and discussion (line 310, to highlight the lack of standardization across the literature).

4) SUD (substance use disorder) is suddenly introduced as a key word, and it is basically a general term.

Subscription, or any relevant medical history is not explored.

Outcomes is a general term. In Pharmacology it is not valid term. Could be just opioid overdose, or opioid overdose deaths, opioid overdose management or ....

The terminology is common in all reports above, and authors approach seems irrelevant to the field.

R: We used the keyword SUD (substance use disorder) to be comprehensive and capture as much as we could in the first screening, in case there were papers looking at other types of substance use disorders, but that also included OUDs. We have added this justification in the manuscript (line 101-103).

As with race/ethnicity, we used the original outcomes reported by the papers and did not try to reclassify any outcome. For the sake of transparency, we used the exact same terms used by the original paper. Because some papers reported on deaths and others reported on non-fatal opioid overdoses and emergency medical service activations related to overdoses, we used the term OUD-related outcome to refer to the larger group of outcomes studied. Due to the small number of studies, we decided for including all these outcomes even though they come from different data sources and represent different burden for the population. We described the data sources and outcomes in the tables to facilitate comparison and increase transparence about what we are comparing. We have included an explanation in the manuscript (lines 180-184).

5) The last major issue is the focus in urban areas. Actually, there is none, or a justification of this type of “review”/search combinations, and future policies.

R: We have clarified the justification for the focus on the urban area (lines 115-119).

“We limited the search to urban areas because those were the areas first affected by the pandemic and where much of OUD service delivery happens. In addition, urban areas have a larger share of minoritized populations and represent highly unequal places, while at the same time having policy-making capacity.”

6) I wish I wouldn’t add ethical issues to the list. I am just wondering who would benefit from this work, and why official reports and comparisons have been omitted.

R: Our focus was to synthesize the peer-reviewed literature on disparities in OUD-related outcomes during the pandemic. We did not aim to summarize the equity policies during the pandemic.  We also did not aim to characterize OUD outcomes in general, but among race/ethnic groups in urban areas of the country, with a focus on disparities. We believe that researchers and policy makers would benefit from having summarized information on this issue, especially when it comes to planning future research. For example, researchers would benefit by understanding the importance of examining outcomes stratified by race/ethnicity (instead of using race/ethnicity simply as an “adjustment”). Researchers would also benefit from realizing that inconsistencies in race/ethnicity reporting can lead to difficulty understanding how the various pieces of research on disparities fit together. Policy makers would benefit from this work by better understanding the race/ethnicity gaps in their cities and local communities to inform the design of policies to address these gaps. These gaps are sometimes difficult to study with national-level data as more granular data is needed to identify local needs. Many of the studies we compiled used local data and the variability of results could simply be a reflection of these different needs. We have included a “Policy recommendation” section in the manuscript (lines 280 – 288) to highlight potential uses of the study.

Please find revised manuscript with track changes attached.

Round 2

Reviewer 4 Report

The authors clearly improved the manuscript, therefore, in my opinion, it is now susceptible for publication

This manuscript is a resubmission of an earlier submission. The following is a list of the peer review reports and author responses from that submission.